# Quality of life of the cancer patients receiving home-based palliative care in Dhaka city of Bangladesh

Jheelam Biswas[1,2]*, Mithila Faruque[1], Palash Chandra Banik[1], Nezamuddin Ahmad[2], Saidur Rahman Mashreky[1]

1 Department of Noncommunicable Diseases, Bangladesh University of Health Sciences (BUHS), Dhaka, Bangladesh, 2 Department of Palliative Medicine, Bangabandhu Sheikh Mujib Medical University (BSMMU), Dhaka, Bangladesh

* jheelam.biswas@gmail.com

**Data Availability Statement:** All data relevant to the study are accessible in the Mendeley database at DOI: 10.17632/d4jzhzxjzj.1 (https://data.mendeley.com/datasets/94jg7jkt6n/1).

## Abstract

### Background

The concept of home-based palliative care has been recently introduced in Bangladesh, but the patients' quality of life remains unexplored. This study aimed to assess the quality of life and its determinants of the cancer patients receiving home-based palliative care in Dhaka, Bangladesh.

### Methods

This cross-sectional study was conducted among 51 surviving cancer patients above 18 years registered under the home-based care service of the Department of Palliative Medicine, Bangabandhu Sheikh Mujib Medical University, Dhaka, Bangladesh. Data was collected by face-to-face interview using a structured questionnaire based on the "Functional Assessment of Chronic Illness Therapy-Palliative (FACIT-Pal)" questionnaire from February to March 2019. Descriptive analysis was done for the socio-demographic, disease and treatment related factors. Mann-Whiteney U test, Kruskal-Wallis H test, and logistic regression were done to determine the relationships between independent variables and QoL.

### Result

The majority of the patients (76.5%) were women. The mean age of the respondents was 56.2±4.8 years. Common primary sites of cancer were breast (39.2%), gastrointestinal (17.6%), and genitourinary system (23.5%). The median duration of getting home-based care was four months. The most prevalent problems were pain, sadness, feeling ill, and lack of satisfaction regarding sexual life. The majority (88.2%) of the patients had an average and above-average quality of life. Although, 92.1%patients had average or above-average social and emotional wellbeing, 60.8% had below-average physical wellbeing. Patients' marital status, belief about disease prognosis, and duration of getting home-based care had a positive influence, and age negatively influenced the quality of life.

**Funding:** The authors received no specific funding for this work.

**Competing interests:** The authors have declared that no competing interests exist.

## Conclusion

The majority of the patients receiving home-based palliative care in Dhaka city had average or above-average quality of life. However, these patients had better social and emotional wellbeing, but the physical wellbeing and symptom control were below-average according to the individual domain.

## Background

Cancer is the 2nd leading cause of death globally, and 70% of these deaths occur in low and middle-income countries [1]. Cancer itself and its associated psychosocial and spiritual problems lower the patients' quality of life significantly [2]. Quality of life of cancer patients is a multidimensional concept consisting of symptom management, physical, psychological, social, and spiritual wellbeing [3]. The holistic approach of palliative care can improve these patients' and their families' quality of life by ensuring a peaceful course through the illness, dealing with the terminal stage of life, and a dignified death [4]. Worldwide, about 40 million people need palliative care, and 34% of them are diagnosed with cancer of different stages, but only about 14% of them are currently receiving palliative care [5].

Among different palliative care delivery system models, the cost-effectiveness and higher patient satisfaction reported in home-based palliative care services increased the popularity of this service around the world [6]. It has been evidenced in several studies that cancer patients receiving home-based palliative care have higher physical performance scores and less depression and anxiety [7–9].

In Bangladesh, the concept of palliative care is still developing. Many patients with different stages of cancer suffering from pain and other symptoms cannot seek institutional care [10]. Although there is no population-based registry of cancer in Bangladesh, it has been estimated that about 0.14 to 0.2 million new patients are being diagnosed with cancer, and cancer causes 10% of annual deaths [11, 12]. Approximately 0.6 million patients need palliative care in Bangladesh, but less than 4,000 people have received this care until now [13, 14].

There is no exact data regarding the number of patients in need of home-based care in Bangladesh. Palliative care is being delivered to a limited extent by two governmental and eight private organizations. Still, these services are negligible in contrast to the enormous unmet need for palliative care in Bangladesh. Also, due to a lack of proper recordkeeping and collaboration, the extent of their services remained unexplored. The Department of Palliative Medicine of Bangabandhu Sheikh Mujib Medical University has taken a pioneering role in this field since 2007 delivering home-based palliative care since 2008, with 643 patients being enlisted to palliative care to date [15]. However, no study has explored the quality of life of these patients receiving this service. This study will assess the quality of life of the cancer patients receiving home-based palliative care in Bangladesh and the factors affecting them.

## Methods

### Study design and setting

This cross-sectional study was conducted amongst all surviving cancer patients currently registered under the Department of Palliative Medicine, Bangabandhu Sheikh Mujib Medical University, Shahbag, Dhaka, which is one of the leading medical universities of Bangladesh and one of the first institutions providing home-based palliative care in this country. Data collection was carried out in February and March 2019.

## Home-based palliative care

All the cancer patients under this study received home-based palliative care provided by the Department of Palliative Medicine, Bangabandhu Sheikh Mujib Medical University, along with or without concomitant cancer treatment. The home care team consists of 1 doctor, 2–3 nurses, 2–3 trained palliative care assistants (PCA). The home care team works six days per week. Patients with a Palliative Performance Score (PPS) below 50 (which includes ambulation, activity and evidence of the disease, self-care, intake, and consciousness level of the patients and scored based on observer's assessment) are usually considered eligible for this service. But those who are unable to attend follow-up in hospital (due to distance, financial constraints, or lack of caregivers) irrespective of their disease stage are also included in this service. This service includes wound care, adjusting medications, general physical and mental care and follow-up, as well as interaction with the caregivers to give temporary respite from their care giving duties. The PCAs do the initial visits. They are specially trained individuals involved in the initial need assessment of the patients, minor wound care, general physical care, helping the family caregivers, and listening to the patients' and their caregivers' problems. They note their assessments in a structured format, and inform doctors and nurses. Based on their initial assessment priority and frequency of the visits are determined. Usually, every patient gets 2–3 visits per month, although extra visits are given based on patients' condition and caregiver demand.

## Sample criteria

All the surviving cancer patients registered under this service up to February 2019, above 18 years of age and willing to participate were included in the study. Those who were delirious, disoriented, or unable to communicate were excluded. Those caregivers (paid or family members) who take care of the patients at least 5 days per week are included in the study. Occasional caregivers were excluded.

## Sample size

According to the Center of Palliative Care (CPC) database up to February 2019, the number of registered cancer patients receiving home-based palliative care was 60. During data collection 3 patients died, 4 patients were not eligible for the study due to delirium, and 3 patients refused to give informed consent. So the final sample size of the study was 51.

## Data collection procedure

Data was collected by the investigator and accompanying home care team using a structured questionnaire in two parts. The first part contained the socio-demographic, disease, treatment, and primary caregiver-related information collected from the hospital record.

The second part contained a Bangla version of "Functional Assessment of Chronic Illness Therapy-Palliative (FACIT-Pal)" (version 4) questionnaire used after obtaining permission from the FACIT group. This version of FACIT-Pal was translated and linguistically validated based on the methodology developed by Eremenco S et al 2005 by the FaCIT team [16]. This contained questions regarding physical, social, emotional, and functional wellbeing. This part of data was collected through face-to-face interviews with the patients and their primary caregivers (family members or paid caregivers).

The investigators accompanied the home care team to the patients' home, and the interviews were conducted in their presence. One of the major concerns during the study was the breakdown of the patient during the administration of the questionnaire. Whenever the

patient appeared to be at the point of breakdown, the interview was stopped and the help of the palliative care team sought to support the patient.

The informed consent was obtained from both the patients and their primary caregivers. Mini Mental State Examination (MMSE) was done to determine the consent giving capacity of the patient. The consent was obtained either in written or verbal form depending on the patients' physical condition.

The patients and the caregivers were recruited in pair and were interviewed together. Whenever a patient is unable to response to any question verbally, the answer was obtained from the caregiver with the permission from the patient. Also, some of the very frail patients had difficulty in communicating directly with the investigators. In such cases help from the caregivers was taken in explaining the questions to the patients, and obtaining their answers. Sensitive questions were asked privately, also was allowed to write down (if a patient was uncomfortable to discuss openly). The duration of each interview was 30 minutes to 1 hour. Two to three patients were interviewed each day. Very frail patients were given multiple visits to complete an interview.

### Data analysis

Conversion of FACIT-Pal quality of life score was done using the FACIT-Pal administration guideline in Microsoft Excel 2010 and entered in SPSS version 22.0, editing and logical checking was done and analyzed.

Categorical variables such as sex, education, marital and occupational status, knowledge and belief about disease prognosis, treatment and their side effects, the relationship of the primary caregiver with the patient were reported as frequency and percentage. Continuous variables such as age, monthly family income, duration of getting home-based palliative care were presented in mean, SD, and median as appropriate.

Quality of life was categorized into three categories by mean±1SD. Descriptive statistics were used to describe the quality of life,and its sub-domains and the quality of life index score was presented in mean and SD. The value below the lower limit of mean-1SD was categorized as below average, the range between an upper and lower limit of mean±1SD was categorized as average, and the value above mean+1SD was categorized as the above-average quality of life.

Mann-Whitney U test and Kruskal-Wallis H test were done to see the relationship among sub-domains of quality of life and different socio-demographic data, disease, treatment-related factors, and symptom profile.

Multiple and binary logistic regression analysis was done to determine the predictors of quality of life (age, marital status, duration of getting home-based care, belief about prognosis) among the study subjects.

### Ethical considerations

Ethical approval for both the research and consent procedure (Approval no: BUHS/BIO/EA/ 18/158, date:18/10/2018) was obtained from the Ethical Review Committee, Bangladesh University of Health Sciences, and permission for data collection was obtained from the Department of Palliative Medicine, Bangabandhu Sheikh Mujib Medical University. The written informed consent was taken from all the eligible patients and their primary caregivers. Sensitive questions were discussed privately. As they were terminally ill patients, their health conditions were considered during data collection.

### Results

The majority (76.5%) of the patients was women, and the mean age was 56.25±14.8 years. More than half (58.8%) of the patients were married and lived with their partners. Almost 97%

of the patients had family members as their primary caregivers, mostly their children (53.2%) or spouses (29.8%), and 57.6% of the primary caregivers were women (Table 1).

The majority (94.1%) of the patients knew that they had cancer, and 72.5% believed that the prognosis of their disease is not good. Common sites of the primary cancer were breast (39.2%), genitourinary system (23.5%), and gastrointestinal tract (17.6%). The most prevalent

**Table 1. Socio-demographic characteristics of the patients and primary caregivers.**

| Variables | n (%) | 95% CI | |
| --- | --- | --- | --- |
| | | Lower bound | Upper bound |
| **Socio-demographic characteristics of the patients (n = 51)** | | | |
| **Sex** | | | |
| Men | 12 (23.5) | 11.9 | 35.1 |
| Women | 39 (76.5) | 64.9 | 88.1 |
| **Age**, *years* | | | |
| *Mean ± SD* | 56.25±14.8 | | |
| <45 | 13(25.5) | 13.5 | 37.5 |
| 45–65 | 26 (51.0) | 37.3 | 64.7 |
| >65 | 12 (23.5) | 11.9 | 35.1 |
| **Marital status** | | | |
| Single (unmarried/divorced/widow) | 21 (41.2) | 27.7 | 54.7 |
| Married | 30 (58.8) | 45.3 | 72.3 |
| **Educational status** | | | |
| Illiterate | 6 (11.8) | 2.9 | 20.7 |
| Primary | 16 (31.4) | 18.7 | 44.1 |
| Up to higher secondary | 18(35.3) | 22.2 | 48.4 |
| Graduate or above | 11 (21.6) | 10.3 | 32.9 |
| **Occupation before illness** | | | |
| Service holder | 11 (21.6) | 10.3 | 32.9 |
| Home maker | 30 (58.8) | 45.3 | 72.3 |
| Others | 10 (19.6) | 8.7 | 30.5 |
| **Characteristics of primary caregivers (n = 47)**[*] | | | |
| **Age**, *years* | | | |
| Mean±SD | 42.3±16.45 | | |
| <31 | 15 (31.9) | 13.3 | 18.6 |
| 31–50 | 20 (42.6) | 28.5 | 56.7 |
| >50 | 12 (25.5) | 13.0 | 38.0 |
| **Sex** | | | |
| Men | 20 (42.5) | 28.4 | 56.6 |
| Women | 27 (57.4) | 43.3 | 71.5 |
| **Educational Status** | | | |
| Up to primary | 8(17.0) | 6.3 | 27.7 |
| Up to higher secondary | 19 (40.4) | 26.4 | 54.4 |
| Graduate or above | 20 (42.6) | 28.5 | 56.7 |
| **Relationship with the patients** | | | |
| Spouse | 14 (29.8) | 16.7 | 42.9 |
| Children | 25 (53.2) | 38.9 | 67.5 |
| Others | 8(17.0) | 6.3 | 27.7 |

[*]Four participants had no caregiver

cancers among men were cancers of the genitourinary system (41.7%) and gastrointestinal tract (33.3%). Among women, the most prevalent cancers were carcinoma of the breast (51.3%), genitor-urinary system (17.9%), and gastrointestinal tract (12.8%) (S1 Table). More than half (55.8%) of the patients had metastasis at the time of referral to palliative care, and 80% of them were currently only on palliative management. The median duration of receiving home-based palliative care of the patients was four months (ranging from 6 days to 1 year) (Table 2).

The majority (88.2%) of the patients had an average or above-average quality of life. However, when we observed the sub-domains of quality of life (social, emotional), we found that 92.1% of the patients had average or above-average social and emotional wellbeing. Still, no one reported above average in the physical domain and mostly had below average (60.8%) physical wellbeing (Table 3).

A significant relationship (p<0.05) was found among median scores of social wellbeing in terms of marital status, with married patients having higher social wellbeing and physical wellbeing irrespective of their disease staging. Those who were in the early stage of cancer had higher physical wellbeing. Median score variation was significant in all sub-domains in belief about prognosis and the duration of getting home-based care (Table 4).

The most severe symptoms experienced by the patients were sadness (58.8%), feeling ill (54.9%), fear of death (52.9%), lack of energy (43.1%),pain (47.1%), and loss of hope (31.4%). More than two-thirds (66.6%) of the patients were not satisfied with their sexual life. More than half of the patients (58.8%) had good mental support from the families and could communicate with them. Almost half of them (49%) were not able to do their day to day activities. No significant relationship had been observed between the quality of life and symptom profile (Table 5).

Marital status influenced the quality of life of the patients positively. Those who were married had 4.8 times better quality of life than those who were single. The patients who believed the disease's prognosis was getting worse than before had lower quality of life than those who thought the prognosis was better or the same (Table 6).

The age of the patients had a negative correlation with the quality of life. Longer duration of home-based care had a positive and significant influence on the quality of life (Table 7).

## Discussion

Home-based palliative care has been introduced recently in Bangladesh. This is the first study in Bangladesh assessing the quality of lifeand their determinants of the cancer patients' receiving such care.

The majority (88.2%) of the patients in this study had average or above the quality of life, which is comparable to cancer patients receiving home-based care in India who mostly reported moderate to a high quality of life [17, 18].

When we looked into each sub-domain, our research found that most (60.8%) patients had below-average physical wellbeing. The prevalence of severe pain (43.1%) was very high in our study despite getting regular home-based care. In this study, more than half of the patients reported other symptoms such as severe lack of energy, feeling ill, and nausea even after getting regular home-based care for the median duration of 4 months. It indicates poor symptom control of the patients in our study. In two separate studies, it was evidenced that home-based palliative care improves multiple symptoms, including pain, nausea, and fatigue, within ten weeks of receiving palliative care [8, 19]. This indicates a major lack in symptom control by the home care team. The exact reasons behind the poor symptom control and physical wellbeing were not explored in our study, however the few home-care visits per patient, lack of proper

**Table 2. Disease and treatment related factors (n = 51).**

| Variables | n (%) | 95% CI | |
|---|---|---|---|
| | | Lower bound | Upper bound |
| **Acknowledged that disease is cancer** | | | |
| Yes | 48 (94.1) | 87.6 | 100.6 |
| No | 03 (5.9) | 0.0 | 12.4 |
| **Belief about prognosis** | | | |
| Better or same as before | 14(27.5) | 15.2 | 39.8 |
| Worse than before | 37 (72.5) | 60.2 | 84.8 |
| **Disfigurement** | | | |
| Mild | 11 (21.6) | 10.3 | 32.9 |
| Moderate | 16 (31.4) | 18.7 | 44.1 |
| Severe | 7 (13.7) | 4.3 | 23.1 |
| **Primary sites of cancer** | | | |
| Gastrointestinal system | 9 (17.6) | 7.1 | 28.1 |
| Genitourinary system | 12 (23.5) | 11.9 | 35.1 |
| Breast | 20 (39.2) | 25.8 | 52.6 |
| Others | 10 (5.6) | 0.0 | 11.9 |
| **Presence of metastasis at referral** | | | |
| Yes | 29 (55.8) | 42.2 | 69.4 |
| No | 22 (42.3) | 28.7 | 55.9 |
| **Staging of cancer at referral** | | | |
| Up to stage III | 12 (23.1) | 11.5 | 34.7 |
| Stage IV | 25 (48.1) | 34.4 | 61.8 |
| Unknown | 15 (28.8) | 16.4 | 41.2 |
| **Co-morbidities** | | | |
| DM | 11(6.1) | 0.0 | 12.7 |
| Cardiovascular | 13 (7.2) | 0.1 | 14.3 |
| COPD | 1 (0.6) | 0.0 | 2.7 |
| Nil | 26 (86.2) | 76.7 | 95.7 |
| **Current treatment** | | | |
| Chemotherapy along with palliative management | 9 (17.3) | 6.9 | 27.7 |
| Only palliative management | 42 (80.7) | 69.9 | 91.5 |
| **Presence of side effects*** | | | |
| Yes | 13 (25.0) | 13.1 | 36.9 |
| No | 38 (78.1) | 66.7 | 89.5 |
| **Duration of getting home-based palliative care (months)** | | | |
| <1 | 18 (35.3) | 22.2 | 48.4 |
| 1–6 | 15 (29.4) | 16.9 | 41.9 |
| >6 | 18 (35.3) | 22.2 | 48.4 |
| Median duration | 4 | | |

*nausea, vomiting, constipation, anemia, diahorrea, discoloration of skin, weakness, fever, alopecia

communication between patients and the home care team, lack of necessary skills, insufficient patient and or caregiver adherence, patient education and disease progression might play a role.

Interestingly, despite having poor symptom control and below-average physical wellbeing, 35.5% of the patients were self-dependent, which indicated better control of the disease [20].

**Table 3. Quality of life (QoL) of the respondents and domain-wise distribution (n = 51).**

| Domain | Mean±SD | Categories | n (%) |
|---|---|---|---|
| **Total QoL** | 90.1±34.5 | Below average (<55.6) | 6 (11.8) |
| | | Average (55.6–124.6) | 35 (68.6) |
| | | Above average (>124.6) | 10 (19.6) |
| *Sub-domains*: | | | |
| **Physical wellbeing** | 12.7±7 | Below average (<5.7) | 31 (60.8) |
| | | Average (5.7–19.7) | 20 (39.2) |
| **Social wellbeing** | 17.06±5.8 | Below average (<11.26) | 5 (9.8) |
| | | Average (11.26–22.86) | 37 (72.5) |
| | | Above Average (>22.86) | 9 (17.6) |
| **Emotional wellbeing** | 10.1±6.7 | Below average (<3.4) | 4 (9.8) |
| | | Average (3.4–16.8) | 38 (72.5) |
| | | Above average (>16.8) | 9 (17.6) |
| **Functional wellbeing** | 11.9±6.7 | Below average (<5.2) | 6 (11.8) |
| | | Average (5.2–18.6) | 35 (68.6) |
| | | Above average (>18.6) | 10 (19.6) |

Quality of life (QoL) and its subdomains were categorized to below average, average and above average by Mean ±1SD

Several studies found a significant relationship between self-dependency and quality of life, although no relationship between these two variables was found in this study [21]. It had also been observed by Peters and Sellick that patients getting home-based care had better symptom control and self-dependency than those receiving institution-based care [7]. Nevertheless, our study made no comparison between home-based and institution-based care.

**Table 4. Relationship of sub-domains of quality of life (QoL) with different variables (n = 51).**

| Variables | PWB Mean rank | *p*-value | SWB Mean rank | *p*-value | EWB Mean rank | *p*-value | FWB Mean rank | *p*-value |
|---|---|---|---|---|---|---|---|---|
| **Marital status**** | | | | | | | | |
| Single | 16.5 | 0.94 | 7.3 | **0.04** | 17.0 | 1.0 | 12.6 | 0.4 |
| Married | 17.05 | | 17.9 | | 17.0 | | 17.4 | |
| **Belief about Prognosis*** | | | | | | | | |
| Better or same | 38.2 | **0.01** | 33.0 | **0.02** | 39.08 | **0.00** | 37.5 | **0.00** |
| Worse | 19.3 | | 20.2 | | 19.2 | | 19.4 | |
| **Staging**** | | | | | | | | |
| Up to stage III | 35.2 | **0.04** | 22.7 | 0.44 | 28.1 | 0.66 | 30.5 | 0.24 |
| Stage IV | 20.6 | | 27.8 | | 24.2 | | 22.7 | |
| Unknown | 28.6 | | 25.2 | | 27.7 | | 28.7 | |
| **Duration of home-based palliative care**** | | | | | | | | |
| <1 month | 15.7 | **0.00** | 21.5 | **0.04** | 21.0 | **0.03** | 17.6 | **0.00** |
| 1–6 months | 26.5 | | 29.5 | | 23.1 | | 24.8 | |
| >6 months | 35.7 | | 27.5 | | 33.3 | | 35.3 | |

*Mann-Whitney U test done;

**Kruskal-Wallis H test done;

#Higher score indicates better quality of life;

PWB = Physical wellbeing; EWB = Emotional wellbeing; FWB = Functional wellbeing; SWB = Social wellbeing

**Table 5. Symptom profile of each domain of quality of life (n = 51).**

| Variables | Not at all | A little bit | Some-what | Quite a bit | Very much |
|---|---|---|---|---|---|
| | | | n (%) | | |
| **Physical** | | | | | |
| Lack of energy | 7 (13.7) | 6 (11.8) | 7 (13.7) | 7 (13.7) | 24 (47.1) |
| Nausea | 21 (41.2) | 11 (21.6) | 5 (9.8) | 5 (9.8) | 9 (17.6) |
| Pain | 5 (9.8) | 10 (19.6) | 5 (9.8) | 9 (17.6) | 22 (43.1) |
| Feeling ill | 9 (17.6) | 7 (13.7) | 2 (3.9) | 5 (9.8) | 28 (54.9) |
| **Social** | | | | | |
| Family support | 4 (7.8) | 0 (0.0) | 5 (9.8) | 12 (23.5) | 30 (58.8) |
| Peer support | 29 (56.9) | 6 (11.8) | 6 (11.8) | 7 (13.7) | 3 (5.9) |
| Sex life | 34 (66.7) | 4 (7.8) | 2 (3.9) | 5 (9.8) | 6 (11.8) |
| **Emotional** | | | | | |
| Sadness | 2 (3.9) | 9 (17.6) | 3 (5.9) | 7 (13.7) | 30 (58.8) |
| Loss of hope | 13 (25.5) | 9 (17.6) | 4 (7.8) | 9 (17.6) | 16 (31.4) |
| Anxiety | 6 (11.8) | 10 (19.6) | 9 (17.6) | 14 (27.6) | 12 (23.5) |
| Fear of death | 9 (17.6) | 7 (13.7) | 4 (7.8) | 4 (7.8) | 27 (52.9) |
| **Functional** | | | | | |
| Ability to do daily work | 25 (49.0) | 9 (17.6) | 3 (5.9) | 4 (7.8) | 10 (19.6) |
| Good sleep | 1 (2.0) | 22 (43.1) | 12 (23.5) | 5 (9.8) | 11 (21.6) |
| Enjoying life | 20 (39.2) | 8 (15.7) | 13 (25.5) | 3 (5.9) | 7 (13.7) |
| **Additional** | | | | | |
| Communication with peers | 28 (54.9) | 7 (13.7) | 3 (5.9) | 9 (17.6) | 4 (7.8) |
| Constipation | 15 (29.4) | 6 (11.8) | 7 (13.7) | 9 (17.6) | 14 (27.5) |
| Weight loss | 16 (31.4) | 12 (23.5) | 5 (9.8) | 10 (19.6) | 8 (15.7) |
| Vomiting | 25 (49.0) | 13 (25.5) | 5 (9.8) | 5 (9.8) | 3 (5.9) |
| Edema | 17 (33.3) | 13 (25.5) | 5 (9.8) | 5 (9.8) | 11 (21.6) |
| Self dependency | 12 (23.5) | 15 (29.4) | 3 (5.9) | 3 (5.9) | 18 (35.3) |
| Peace of mind | 18 (35.4) | 9 (17.6) | 4 (7.8) | 8 (15.7) | 12 (23.5) |
| Hopefulness | 22 (43.1) | 11 (21.6) | 4 (7.8) | 6 (11.8) | 8 (15.7) |
| Reconciliation with others | 10 (19.6) | 12 (23.5) | 10 (19.6) | 8 (15.7) | 11 (21.6) |
| Communication with family | 14 (27.5) | 8 (15.7) | 5 (9.8) | 8 (15.7) | 16 (31.4) |

**Table 6. Predictors of quality of life (QoL) among cancer patients considering the below average as reference (n = 51).**

| Factors | B | p-value | OR | 95% CI for OR | |
|---|---|---|---|---|---|
| | | | | Lower limit | Upper limit |
| **Marital status** | | | | | |
| Single | Reference | | | | |
| Married | 1.58 | 0.09 | 4.88 | 0.76 | 30.94 |
| **Duration of home-based palliative care** | | | | | |
| >6 months | Reference | | | | |
| <6 months | -3.52 | 0.00 | 0.29 | 0.004 | 0.21 |
| **Belief about disease prognosis** | | | | | |
| Better than before | Reference | | | | |
| Worse than before | -3.66 | 0.02 | 0.026 | 0.003 | 0.26 |

Binary logistic regression was done

**Table 7. Predictors of QoL among cancer patients (n = 51).**

| Variables | Standardized Coefficient β | *p*-value* |
|---|---|---|
| Age | -0.27 (-0.86 to 0.32) | 0.359 |
| Duration of getting home based palliative care (months) | 0.559 (0.127 to 0.992) | 0.012 |

Multiple linear regression analysis (adjusted) was computed and all the precisions were estimated at 95% confidence interval

*The standardized coefficient β was statistically significant at a threshold of *p* < 0.05

Despite having poor physical wellbeing, most of the patients in this study had an average to above-average social and emotional wellbeing. Our finding is similar to the cancer patients' social and emotional wellbeing receiving home-based care in India [16]. However, the reasons behind the better psychosocial wellbeing were not explored in our study. Although studies have shown that home care team visits psychologically boost the patients and make them feel valued [22, 23]. Also, our country's social and family structure might contribute to these patients' psychosocial wellbeing. More than 90% of the patients in our study lived with their families. The majority (82%) of them were taken care of by family members such as children or spouses and they receive psychological support from them, which is very important for social wellbeing, and very common in social structures such as in Bangladesh and India [24]. The majority (58.8%) of the patients received support from their families, which they found very important in going through their illness. Alongside previous studies in Australia and Iran, the patients in our study living with their spouses had a better quality of life [9, 25]. Two-third (66.7%) of these patients were not satisfied with their sexual life, although they had a better quality of life. In two separate studies, it has been evidenced that sexual and marital satisfaction has a positive influence on the quality of life, however our study found no such relationship [25, 26].

Almost half of the patients reported feeling sad most of the time, despite having average to above-average social and emotional wellbeing. This percentage is nearly double compared to the patients in Australia and India receiving home-based care [16, 27]. Our study did not clinically define the feeling of sadness as depression.

In our study longer duration (>6 months) of receiving home-based palliative care had a positive influence on the total quality of life, which is similar to the Zimmermann study where a significant improvement in the quality of life of advanced cancer patients was found after four months of home-based care [28]. In that study, the effect of home-based care for the initial three months had a similar outcome to usual cancer care [28]. Nevertheless, most of the patients who enrolled for home-based care have a limited life expectancy, so the time for palliative care has to make an impact is significantly less.

A contradictory finding in the current study is about the patients' belief about disease progression. In our study, those who believed the prognosis of the disease as getting worse than before had lower quality of life than those who thought the prognosis was better or the same as before. In several studies, it has been found that conspiracy of silence and false hope lead the patients seeking aggressive but non-beneficial treatments. Both of these have a negative consequence on the patients' quality of life [29–31]. We did not explore the exact reason behind this contradiction.

Regarding utilization of Home-based palliative care, most (76.5%) of this service recipients were women. Several studies found that an equal number of men and women seek home-based palliative care in most countries [32]. The possible reasons behind this large number of women seeking Home-based care in this study are the socio-economic and religious structure

of our country, occupation of the patients (58.8% of the patients were homemakers), and the sex of the primary caregiver (57.6% of the primary caregivers were women). In our socio-economic and religious structure, women mostly stay at home and are dependent on their male family members, so they often hesitate to seek care at hospitals situated far from home. However, this study was only restricted to the patients registered under Bangabandhu Sheikh Mujib Medical University, so this study's findings cannot be generalized.

## Conclusion

Although the overall quality of life of the majority of the cancer patients receiving home-based care in this study was average or above average and had better psychosocial wellbeing, their physical wellbeing was not satisfactory. Further research is needed to identify the factors that may influence physical wellbeing and symptom control by the home care team and provide scopes for improving the care to meet patients with advanced cancer and their families.

## Supporting information

**S1 Table. Cancer according to gender (n = 51).**
(DOCX)

## Acknowledgments

The authors gratefully acknowledge the contribution of the home care team members of BSMMU and their coordinator Mridul Sarker and Fazle Noor Biswas to provide information and data collection. The authors also thank Professor Mostafa Zaman, Adjunct Professor, Department of NCD, BUHS, for his valuable opinion for overall research.

## Author Contributions

**Conceptualization:** Jheelam Biswas, Mithila Faruque, Palash Chandra Banik.

**Data curation:** Jheelam Biswas, Mithila Faruque, Palash Chandra Banik.

**Formal analysis:** Jheelam Biswas, Mithila Faruque.

**Investigation:** Jheelam Biswas, Mithila Faruque, Palash Chandra Banik.

**Methodology:** Jheelam Biswas, Palash Chandra Banik.

**Project administration:** Jheelam Biswas.

**Resources:** Jheelam Biswas.

**Software:** Jheelam Biswas, Palash Chandra Banik.

**Supervision:** Mithila Faruque, Palash Chandra Banik, Nezamuddin Ahmad, Saidur Rahman Mashreky.

**Visualization:** Jheelam Biswas, Mithila Faruque, Palash Chandra Banik.

**Writing – original draft:** Jheelam Biswas, Mithila Faruque, Palash Chandra Banik.

**Writing – review & editing:** Mithila Faruque, Palash Chandra Banik, Nezamuddin Ahmad, Saidur Rahman Mashreky.

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
