## [Decision Letter · Decision Letter 0]

16 Feb 2022

PONE-D-21-40845Quality of Life of the Cancer Patients Receiving Home-based Palliative Care in Dhaka City of BangladeshPLOS ONE

Dear Dr. Biswas,

Thank you for submitting your manuscript to PLOS ONE. After careful consideration, we feel that it has merit but does not fully meet PLOS ONE’s publication criteria as it currently stands. Therefore, we invite you to submit a revised version of the manuscript that addresses the points raised during the review process.

We look forward to receiving your revised manuscript.

Kind regards,

Tai-Heng Chen, M.D.

Academic Editor

PLOS ONE

Journal Requirements:

2.  Please describe in your methods section how capacity to provide consent was determined for the participants in this study. Please also state whether your ethics committee or IRB approved this consent procedure. If you did not assess capacity to consent please briefly outline why this was not necessary in this case.

(This study received no specific grant from any funding agency in the public, commercial, or not-for-profit sectors.)

Reviewers' comments:

Reviewer's Responses to Questions

**Comments to the Author**

1. Is the manuscript technically sound, and do the data support the conclusions?

Reviewer #2: Yes

Reviewer #3: Yes

Reviewer #4: Partly

2. Has the statistical analysis been performed appropriately and rigorously? 

Reviewer #2: I Don't Know

Reviewer #3: Yes

Reviewer #4: Yes

3. Have the authors made all data underlying the findings in their manuscript fully available?

Reviewer #2: Yes

Reviewer #3: Yes

Reviewer #4: Yes

4. Is the manuscript presented in an intelligible fashion and written in standard English?

Reviewer #2: Yes

Reviewer #3: No

Reviewer #4: Yes

5. Review Comments to the Author

Reviewer #2: Congratulations on your work. Some suggestions follow, which I hope you will find useful.

= Comments =

114 - Could you elaborate on what palliative care assistants do? In some countries the term "assistant" is also used for doctors for example.

115 - On this line it is not clear if every is patient visited 6 times per week or if visits are planned on a per-need basis. You clarify this in line 121 but I would suggest rewriting in the sense of "the home care team works during 6 days per week".

123 - Data was collected between February and March 2019, has no patient died in that period?

124 - "Those who were delirious, disoriented, or unable to communicate were excluded." Since delirium is a symptom that is frequent in Palliative Care, can you please state how many patients were excluded from all the surviving cancer patients registered?

145 - 150 - I’m having difficulty in understanding this methodology, since you are not comparing QoL with other samples. Therefore, dividing your group of patients in 3 according to mean values seems counterintuitive with calling this groups “below-average”, “average” or “above-average”.

Table 1

. what do the 95% confidence intervals refer to since there is no statistical test mentioned?

. please give a definition for "HSC" (in the first part of the table "higher secondary" is used)

173 - Please define GIT (gastrointestinal tract is used throughout the rest of the text)

Table 2

. what do the 95% confidence intervals refer to since there is no statistical test mentioned?

. the answer “No” for “Acknowledged that disease is cancer” is missing

. “Presence of metastasis during referral” – I would suggest “at referral”; the answer “No” is missing

. “Staging of cancer during referral” - I would suggest “at referral”; stage 3 and 4 should be III and IV if by the AJCC TNM staging system

. “Chemotherapy along with palliative management” – was any patient receiving other cancer treatment, i.e., hormone therapy?

. “Presence of side effects” – is this referring to side effects attributable to chemotherapy? the answer “No” is missing

195 - “in terms of the disease's staging” – Can you please explain what you mean by this?

Table 3

. stage 3 and 4 should be III and IV if by the AJCC TNM staging system

203 - “not satisfied with their partners' sexual relationship” – consider writing “satisfaction regarding their sexual life”, since the current phrase may be interpreted as dissatisfaction with the sexual life of their partners.

Table 5

. consider using “anxiety” instead of “nervousness”

. consider using “edema” instead of “Swelling of body”

276 - 280 - this is a controversial topic. Some studies indicate that knowledge about prognosis at the end-of-life may contribute to promote better communication and avoid situations of conspiracy of silence.

. Lemus-Riscanevo P, Carreño-Moreno S, Arias-Rojas M. Conspiracy of Silence in Palliative Care: A Concept Analysis. Indian J Palliat Care. 2019 Jan-Mar;25(1):24-29.

. Lee H, Ko HJ, Kim AS, Kim SM, Moon H, Choi HI. Effect of Prognosis Awareness on the Survival and Quality of Life of Terminally Ill Cancer Patients: A Prospective Cohort Study. Korean J Fam Med. 2020 Mar;41(2):91-97.

= Typos/grammatical suggestions =

Spaces are missing after full stop/period in multiple sites of the text

47 - "All 51 surviving" - capital letter A

65 - "average or above the quality of life." - or above average quality of life

60 and 65 - "social and emotional wellbeing" vs. "psychosocial" - using the same term in the abstract could be easier to read

79 - these patients and their families' quality of life

93 - "had received" - have received

95 - "two governments" - governmental

113 - "along with or without the definitive cancer treatment." - with or without concomitant cancer treatment

132 - "data were" - although data is plural; consider using "data was" if you agree, since the singular form is more widely used in scientific literature

176 - “during referral” - at the time of referral

285 - “homemaker” – homemakers (plural)

Reviewer #3: This is a really interesting paper and I would recommend it for publication however prior to being published it needs a lot of work with regards to the English and grammar. I have begun to do some initial work on this but prior to publication it needs proper editing

Reviewer #4: Thank you for the opportunity to review this study. The study aim is relevant to the field.

Major comments:

1. Pg 4 line 125-126: The report needs to provide more clarity on the involvement of homecare team in the data collection process to ensure that the study was conducted ethically.

2. Pg 4 line 122: The sample characteristics only specified criteria for patients and the involvement of primary caregivers should have also been defined here. The inclusion and role of the primary caregivers in the study is not clear. How were caregivers approached? Were the patients and caregivers recruited as a pair? How did they manage the ethical issue of pressure to participate for the family caregivers who might not be interested but feel they have to do it because of their patients? How did the family caregivers participate in the interviews? Were they used for proxy reporting for patients or were they interviewed on their own.

3. Pg 4 Line 132-136 and Pg5 line 160-161: The authors stated they obtained Informed consent from either patient or caregiver? Were there patients involved in the study who did not consent to participate and the authors took consent from the caregivers to do proxy reporting on the patients? This raises ethical issues

Minor comments

1. Pg4 line 115: What is palliative performance score? There is a need to provide more information about the data collection tools used in this study. What are their psychometric characteristics, their scoring methods, their lowest and highest obtainable scores, and relevant references. This will help to put the results reported in context.

2. Pg4 line 129-130: What was the procedure for the translation of the FACIT-PAL? What level of validation work has been done on this translated version? Any psychometric properties to be reported? Any reference?

3. Pg5 Line 155: Why were these particular statistical approaches selected to analyse the data? How did the data inform this choice? There is need to justify their relevance.

4. Pg7 line171: This can be presented in clearer unambiguous language

5. Pg line 172-175: As these were not reported in the tables, it is difficult to understand the data as the presented percentages for the two genders did not add up to 100%.

6. pg7 line 177-178: It’s good practice to report the range as well when reporting median

7. Presence of side effects Table 2: The side effects list referred to is not exclusive as it had an 'etc.' This leaves the reader to fill in the gap of what other side effects were reported by the participants. This is dismissive of the patients side effects. This needs to be addressed to ensure all side effects referred to are listed here especially considering that majority of the participants reported below average quality of life on the physical domain

8. Pg9 line 201-202: I think more nuance is needed in reporting this information. What the likert scale measured was the severity of each symptom in individual patients rather than the prevalence ranking of each symptom among all patients. Using most severe symptoms rather than most prevalent would help readers like me understand this better.

9. Table 5: Some of the data on the listed symptoms do not add up to the n of 51. No explanation was provided for this.

10. Pg12 line 241: The authors can consider changing this to adherence. Compliance insinuates a power dynamic that is not encouraged in palliative care as care should be informed by patients’ preferences

11. Pg13 line 279-280: It is important to provide balance here as hoping for better outcomes has also been shown to increase uptake of excessive, overaggressive and unnecessary medical interventions towards the end of life which may reduce quality of life.

6. PLOS authors have the option to publish the peer review history of their article (what does this mean?). If published, this will include your full peer review and any attached files.

Reviewer #2: **Yes: **Michael Luis

Reviewer #3: No

Reviewer #4: No

---

## [Author Response · Author response to Decision Letter 0]

26 Mar 2022

Quality of Life of the Cancer Patients Receiving Home-based Palliative Care in Dhaka City of Bangladesh 

 (Manuscript ID: PONE-D-21-40845) 

Journal requirement comments to the authors-

Comment 1: Please ensure that your manuscript meets PLOS ONE's style requirements, including those for file naming

Reply: Thank you. We have revised the whole manuscript according the journal’s style requirement.

Comment 2: Please describe in your methods section how capacity to provide consent was determined for the participants in this study. Please also state whether your ethics committee or IRB approved this consent procedure. If you did not assess capacity to consent please briefly outline why this was not necessary in this case.

Reply: Thank you for your comments; we have performed Mini Mental State Examination on each patient before obtaining consent. The procedure was approved by Ethical review committee. We have included these statements in our methods section.

Changes in the text: 

Methods:

Data collection procedure: 

Line 156-159: The informed consent was obtained from both the patients and their primary caregivers. Mini Mental State Examination (MMSE) was done to determine the consent giving capacity of the patient. The consent was obtained either in written or verbal form depending on the patients’ physical condition. 

Ethical consideration: 

Line 189-191: Ethical approval for both the research and consent procedure (Approval no: BUHS/BIO/EA/18/158, date:18/10/2018) was obtained from the Ethical Review Committee, Bangladesh University of Health Sciences.

Comment 3: Please clarify the sources of funding (financial or material support) for your study. List the grants or organizations that supported your study, including funding received from your institution. 

Reply: Thank you. We have not received any specific funding from any source. 

Changes in the text: 

Line 348: The authors received no specific funding for this work. 

Reviewer#2’s comments to the authors-

Comment 1: 114 - Could you elaborate on what palliative care assistants do? In some countries the term "assistant" is also used for doctors for example.

Reply: Thank you. The role of palliative care assistants (PCA) are added to the revised manuscript. 

Changes in the text:

Methods: About home based palliative care

Line 123- 127:The PCAs do the initial visits. They are specially trained individuals involved in the initial need assessment of the patients, minor wound care, general physical care, helping the family caregivers,and listening to the patients’ and their caregivers’ problems. They note their assessments in a structured format, and inform doctors and nurses. Based on their initial assessment priority and frequency of the visits are determined.

Comment 2: 115 - On this line it is not clear if every is patient visited 6 times per week or if visits are planned on a per-need basis. You clarify this in line 121 but I would suggest rewriting in the sense of "the home care team works during 6 days per week".

Reply: Thank you. We have corrected the line in the revised manuscript

Changes in the text:

Methods: Home based palliative care

Line 116: The home care team works six days per week.

Comment 3:123 - Data was collected between February and March 2019, has no patient died in that period?

Reply: Thank you for pointing the issue. Among the 60 registered patients 3 patients died during data collection period. They were excluded from the study

Changes in the text:

Methods: Sample size

Line 135-139: According to Center of palliative care (CPC) database up to February 2019, the number of registered cancer patients receiving home based palliative care was 60. During data collection 3 patients died, 4 patients were not eligible for the study due to delirium, and 3 patients refused to give informed consent. So the final sample size of the study was 51.

Comment 4: 124 - "Those who were delirious, disoriented, or unable to communicate were excluded." Since delirium is a symptom that is frequent in Palliative Care, can you please state how many patients were excluded from all the surviving cancer patients registered?

Reply: Thank you for pointing the issue. Among the 60 registered patients 4 patients were excluded from the study due to delirium. 

Changes in the text:

Methods: Sample size

Line 135-139: According to Center of palliative care (CPC) database up to February 2019, the number of registered cancer patients receiving home based palliative care was 60. During data collection 3 patients died, 4 patients were not eligible for the study dueto delirium, and 3 patients refused to give informed consent. So the final sample size of the study was 51.

Comment 5: 145 - 150 - I’m having difficulty in understanding this methodology, since you are not comparing QoL with other samples. Therefore, dividing your group of patients in 3 according to mean values seems counterintuitive with calling this groups “below-average”, “average” or “above-average”.

Reply: Thank you. Yes, we have not compared our study group with any other population. However, we want to see the distribution of QoL for better understanding. 

Comment 6: Table 1

. what do the 95% confidence intervals refer to since there is no statistical test mentioned?

. please give a definition for "HSC" (in the first part of the table "higher secondary" is used)

Reply: Thank you for raising the question. In table 1, 95% CI was mentioned to show the distribution of the sample among the population. No statistical test was done here. 

Also, we have change “HSC” to “upto Higher Secondary” in Table 1. Thank you for pointing out the error. 

Comment 7: 173 - Please define GIT (gastrointestinal tract is used throughout the rest of the text)

Reply: Thank you for pointing out the error. We have replaced “GIT” with “gastrointestinal tract” in the revised manuscript. 

Changes in the text:

Result: 

Line 207: The most prevalent cancers among men were cancers of the genitourinary system (41.7%) and gastrointestinal tract (33.3%).

Comment 8: Table 2

. what do the 95% confidence intervals refer to since there is no statistical test mentioned?

. the answer “No” for “Acknowledged that disease is cancer” is missing

. “Presence of metastasis during referral” – I would suggest “at referral”; the answer “No” is missing

. “Staging of cancer during referral” - I would suggest “at referral”; stage 3 and 4 should be III and IV if by the AJCC TNM staging system

. “Chemotherapy along with palliative management” – was any patient receiving other cancer treatment, i.e., hormone therapy?

. “Presence of side effects” – is this referring to side effects attributable to chemotherapy? the answer “No” is missing

Reply: Thank you for pointing out the errors and raising some important questions. 

1. In table 2, 95% CI was mentioned to show the distribution of the sample among the population. No statistical test was done here.

2. “Presence of metastasis during referral” is replaced with “at referral” in the revised manuscript. Also the answer “No” is added to the table

3. .“Staging of cancer during referral” – is replaced with “at referral” in the revised manuscript. Also ‘stage 3 and 4’ is replaced with ‘stage III and IV’ 

4. During data collection no patient received any other concurrent treatment except chemotherapy along with palliative care

5. The side effects are attributed to any side effects experienced by the patients due to their treatment. These side effects are not only from chemotherapy. But our patients were only receiving chemotherapy during data collection. The answer “No” is added to the revised manuscript.

Comment 9: 195 - “in terms of the disease's staging” – Can you please explain what you mean by this?

Reply: Thank you for mentioning the error. It was a grammatical mistake. We have corrected the line in the revised manuscript.

Changes in the text: 

Result:

Line 230-232: A significant relationship (p<0.05) was found among median scores of social wellbeing in terms of marital status, with married patients having higher social wellbeing and physical wellbeing irrespective of their disease staging.

Comment 10: Table 3: stage 3 and 4 should be III and IV if by the AJCC TNM staging system

Reply: Thank you for mentioning. ‘Stage 3 and 4’ is replaced with ‘stage III and IV’ in the table 3 of the revised manuscript.

Comment 11: 203 - “not satisfied with their partners' sexual relationship” – consider writing “satisfaction regarding their sexual life”, since the current phrase may be interpreted as dissatisfaction with the sexual life of their partners.

Reply: Thank you for mentioning. We have corrected the phrase.

Changes in the text: 

Result:

Line 241: More than two-thirds (66.6%) of the patients were not satisfied with their sexual life.

Comment 12: Table 5

. consider using “anxiety” instead of “nervousness”

. consider using “edema” instead of “Swelling of body”

Reply: Thank you. The terms “Nervousness” and “Swelling of body” has been replaced with “anxiety” and “edema” in table 5 of the revised manuscript. 

Comment 13: 276 - 280 - this is a controversial topic. Some studies indicate that knowledge about prognosis at the end-of-life may contribute to promote better communication and avoid situations of conspiracy of silence.

. Lemus-Riscanevo P, Carreño-Moreno S, Arias-Rojas M. Conspiracy of Silence in Palliative Care: A Concept Analysis. Indian J Palliat Care. 2019 Jan-Mar;25(1):24-29.

. Lee H, Ko HJ, Kim AS, Kim SM, Moon H, Choi HI. Effect of Prognosis Awareness on the Survival and Quality of Life of Terminally Ill Cancer Patients: A Prospective Cohort Study. Korean J Fam Med. 2020 Mar;41(2):91-97.

Reply: Thank you for mentioning the controversy. We have revised our manuscript and found this controversy too. So we have mentioned this controversy in the discussion section, although our study did not explore the exact reason behind the controversy.

Changes in the text:

Discussion: 

Line 316-322: A contradictory finding in the current study is about the patients' belief about disease progression. In our study, those who believed the prognosis of the disease as getting worse than before had lower quality of life than those who thought the prognosis was better or the same as before. In several studies, it has been found that conspiracy of silence and false hope lead the patients seeking aggressive but non beneficial treatments. Both of these have a negative consequence on the patients’ quality of life [30-32]. Although we did not explore the exact reason behind this contradiction.

References: 

30. Lemus-Riscanevo P, Carreño-Moreno S, Arias-Rojas M. Conspiracy of Silence in Palliative Care: A Concept Analysis. Indian J Palliat Care. 2019 Jan-Mar;25(1):24-29.

31. Lee H, Ko HJ, Kim AS, Kim SM, Moon H, Choi HI. Effect of Prognosis Awareness on the Survival and Quality of Life of Terminally Ill Cancer Patients: A Prospective Cohort Study. Koran J Fam Med. 2020 Mar;41(2):91-97.

32. Cardona-Morrell M, Kim J, Turner R, Anstey M, Mitchell I, Hillman K. Non-beneficial treatments in hospital at the end of life: a systematic review on extent of the problem. Eval Health Prof2016;28(4):456-469.

Comment 14: = Typos/grammatical suggestions =

Spaces are missing after full stop/period in multiple sites of the text

47 - "All 51 surviving" - capital letter A

65 - "average or above the quality of life." - or above average quality of life

60 and 65 - "social and emotional wellbeing" vs. "psychosocial" - using the same term in the abstract could be easier to read

79 - these patients and their families' quality of life

93 - "had received" - have received

95 - "two governments" - governmental

113 - "along with or without the definitive cancer treatment." - with or without concomitant cancer treatment

132 - "data were" - although data is plural; consider using "data was" if you agree, since the singular form is more widely used in scientific literature

176 - “during referral” - at the time of referral

285 - “homemaker” – homemakers (plural)

Reply: Thank you for pointing out the mistakes. We have corrected them according to your suggestion.

Changes in the text:

Line 47: This cross-sectional study was conducted among 51 surviving cancer patients

Line 60-61:The majority of the patients receiving home-based palliative care in Dhaka city had average or above average quality of life.

Line 66-67:However, these patients had better social and emotional wellbeing,

Line 78-79: The holistic approach of palliative care can improve these patients' and their families’ quality of life

Line 93: But less than 4,000 people have received this care

Line 96: Palliative care is being delivered to a limited extent by two governmental and eight private organizations.

Line 114: All the cancer patients under the study received home-based palliative care provided by the Department of Palliative Medicine, Bangabandhu Sheikh Mujib Medical University, along with or without concomitant cancer treatment

Line 111: Data collection was carried out in February and March

Line 211: More than half (55.8%) of the patients had metastasis at the time of referral to palliative care,

Line 327-328: 58.8% of the patients were homemakers

Reviewer#3’s comments to the authors-

Comment: This is a really interesting paper and I would recommend it for publication however prior to being published it needs a lot of work with regards to the English and grammar. I have begun to do some initial work on this but prior to publication it needs proper editing

Reply: Thank you. We have tried our best to find out the grammatical mistakes, and correct them in the revised manuscript. 

Reviewer#4’s comments to the authors- 

Major comments:

Comment 1: Pg 4 line 125-126: The report needs to provide more clarity on the involvement of homecare team in the data collection process to ensure that the study was conducted ethically.

Reply: Thank you. We have included the role of home care team in the data collection process in the revised manuscript.

Changes in the text:

Line 151-155: The investigators accompanied the home care team to the patients’ home, and the interviews were conducted in their presence. One of the major concerns during the study was the breakdown of the patient during the administration of the questionnaire. Whenever the patient appeared to be at the point of breakdown, the interview was stopped and the help of palliative care team sought for supporting the patient.

Comment 2: Pg 4 line 122: The sample characteristics only specified criteria for patients and the involvement of primary caregivers should have also been defined here. The inclusion and role of the primary caregivers in the study is not clear. How were caregivers approached? Were the patients and caregivers recruited as a pair? How did they manage the ethical issue of pressure to participate for the family caregivers who might not be interested but feel they have to do it because of their patients? How did the family caregivers participate in the interviews? Were they used for proxy reporting for patients or were they interviewed on their own.

Reply: Thank you for mentioning the issues. We have added the inclusion criteria of the primary caregivers in the methodology section. The primary caregivers were recruited in the interview with the permission of the patients, as they feel comfortable in their presence. Sometimes some of the patients had difficulty to communicate with the investigators directly; some of them use their own language to answer some of the questions. We sought help from the caregiver to explain the questions and interpret the patient’s language. Both patients and their caregivers were recruited as pair and interviewed together. Anyone (patients or caregivers) who refused to take part in the study were excluded. 

Changes in the text:

Methods: Data collection procedure

Line 160-168:The patients and the caregivers were recruited in pair and were interviewed together. Whenever a patient is unable to response to any question verbally, the answer was obtained from the caregiver with the permission from the patient. Also, some of the very frail patients had difficulty in communicating directly with the investigators. In such cases help from the caregivers was taken in explaining the questions to the patients, and obtaining their answers. Sensitive questions were asked privately, also was allowed to write down (if a patient was uncomfortable to discuss openly). Duration of each interview was 30 minutes to 1 hour. Two to three patients were interviewed in each day. Very frail patients were given multiple visits to complete an interview. 

Comment 3: Pg 4 Line 132-136 and Pg5 line 160-161: The authors stated they obtained Informed consent from either patient or caregiver? Were there patients involved in the study who did not consent to participate and the authors took consent from the caregivers to do proxy reporting on the patients? This raises ethical issues

Reply: Thank you for mentioning the issue. We have taken informed consent from both patient and their caregivers together. Caregiver consent was taken after obtaining consent from the patient. If a patient refused to give consent no proxy reporting from the caregiver was not taken.

Changes in the text:

Methods: Data collection procedure

Line 156-159: The informed consent was obtained from both the patients and their primary caregivers. Mini Mental State Examination (MMSE) was done to determine the consent giving capacity of the patient. The consent was obtained either in written or verbal form depending on the patients’ physical condition. 

Minor comments:

Comment 1: Pg 4 line 115: What is palliative performance score? There is a need to provide more information about the data collection tools used in this study. What are their psychometric characteristics, their scoring methods, their lowest and highest obtainable scores, and relevant references. This will help to put the results reported in context.

Reply: Thank you for mentioning the queries. 

Palliative Performance Score (PPS) is an observer based scoring process to determine the possible life span of the terminally ill patients. It has 5 components- ambulation, evidence and extent of the disease, self-care, intake and consciousness level. However, this scoring system is used by the home care team to determine the eligibility of the patients for this service. We did not use this tool in our study. 

We have only used FaCIT-Pal, which has psychometric properties based on four sub-scales- Physical, emotional, social and additional. The tool is developed and validated by FaCIT Team (https://www.facit.org). The administration guideline and the scoring system in detail are added in the supplementary file S3: FaCIT-Pal administration guideline. 

Changes in the text:

Methods: Home based palliative care

Line 116-119: Patients with a Palliative Performance Score (PPS) below 50 (which includes ambulation, evidence and extent of the disease, self-care, intake, and consciousness level of the patients and scored based on observer’s assessment) are usually considered eligible for this service.

Methods: Data collection procedure

Line 144-148: The second part contained a Bangla version of “Functional Assessment of Chronic Illness Therapy-Palliative (FACIT-Pal)" (version 4) questionnaire used after obtaining permission from the FACIT group. This version of FACIT-Pal was translated and linguistically validated based on the methodology developed by Eremenco S et al 2005 by the FaCIT team[29]. This contained questions regarding physical, social, emotional, and functional wellbeing. 

Data analysis:

Line 170: Conversion of FACIT-Pal quality of life score was done using the FACIT-Pal administration guideline (supplementary file S3)

Comment 2: Pg 4 line 129-130: What was the procedure for the translation of the FACIT-PAL? What level of validation work has been done on this translated version? Any psychometric properties to be reported? Any reference?

Reply: Thank you. The Bengali FACIT-Pal was translated and linguistically validated based on the methodology developed by Eremenco S et al 2005 by the FaCIT team (https://www.facit.org) (1). The psychometric properties of FaCIT-Pal have been already reported by Shinall MC et al 2018 (2). Although the formal assessment of the psychometric properties of Bangla version has not been completed yet, but the data is comparable with the English version because of the methodology followed in the translation. 

1. EremencoS ,Cella D, Arnold B. A Comprehensive Method for the Translation and Cross-Cultural Validation of Health Status Questionnaires. Eval Health Prof. 2005;28(2):212-232

2. Shinall MC, Ely EW, Karlekar M, Robbins SG, Chandrasekhar R, Martin SF. Psychometric Properties of the FACIT-Pal 14 Administered in an Outpatient Palliative Care Clinic. Am J Hosp Palliat Care. 2018 Oct;35(10):1292-1294. doi: 10.1177/1049909118763793.

Changes in the text:

Methods: Data collection procedure

Line 144-148: The second part contained a Bangla version of “Functional Assessment of Chronic Illness Therapy-Palliative (FACIT-Pal)" (version 4) questionnaire used after obtaining permission from the FACIT group. This version of FACIT-Pal was translated and linguistically validated based on the methodology developed by Eremenco S et al. 2005 by the FaCIT team [29].

Comment 3: Pg5 Line 155: Why were these particular statistical approaches selected to analyse the data? How did the data inform this choice? There is need to justify their relevance.

Reply: Thank you for asking. Logistic regression was done to determine the predictors affecting the study subjects’ quality of life. As the data for ‘Marital status, and belief about prognosis’ were binominal, so we selected Binary logistic regression. As the data for ‘Age’ is multinominal, so multiple logistic regression was chosen. As duration of getting palliative care is a continuous data, we have at first used multiple logistic regression. But for determining which duration is affecting the QoL, we have re-grouped the data in “less than 6 months and more than 6 months”, and used binary logistic regression. 

Comment 4: Pg7 line171: This can be presented in clearer unambiguous language

Reply: Thank you. We have corrected the line. 

Changes in the text: Result

Line 205-206: The majority (94.1%) of the patients knew that they had cancer, and 72.5% believed that the prognosis of their disease is not good.

Comment 5: Pg line 172-175: As these were not reported in the tables, it is difficult to understand the data as the presented percentages for the two genders did not add up to 100%.

Reply: Thank you. We have added this data in the supplementary table- S1 Table: Cancer according to gender (n=51)

Comment 6: pg 7 line 177-178: It’s good practice to report the range as well when reporting median

Reply:Thank you. We have added the range of duration of getting palliative care of the patients in the revised manuscript.

Changes in the text: Result

Line 213-214: The median duration of receiving home-based palliative care of the patients was four months (ranged from 6 days to 1 year) 

Comment 7: Presence of side effects Table 2: The side effects list referred to is not exclusive as it had an 'etc.' This leaves the reader to fill in the gap of what other side effects were reported by the participants. This is dismissive of the patients side effects. This needs to be addressed to ensure all side effects referred to are listed here especially considering that majority of the participants reported below average quality of life on the physical domain

Reply: Thank you. We have added all the side effects reported by the patients in the footnote of Table 2. 

Changes in the text: Result

Table 2 footnote: *nausea, vomiting, constipation, anemia, diahorrea, discoloration of skin, weakness, fever, alopecia

Comment 8: Pg9 line 201-202: I think more nuance is needed in reporting this information. What the likert scale measured was the severity of each symptom in individual patients rather than the prevalence ranking of each symptom among all patients. Using most severe symptoms rather than most prevalent would help readers like me understand this better.

Reply: Thank you. We have arranged the symptoms according to severity in the revised manuscript.

Changes in the text: Result

Line 239-240: The most severe symptoms experienced by the patients were sadness (58.8%), feeling ill (54.9%), fear of death (52.9%) ,lack of energy (43.1%), pain (47.1%), and loss of hope (31.4).

Comment 9: Table 5: Some of the data on the listed symptoms do not add up to the n of 51. No explanation was provided for this.

Reply: Thank you for pointing out the errors. We have re-analyzed our data and corrected the errors in Table 5. 

Comment 10: Pg12 line 241: The authors can consider changing this to adherence. Compliance insinuates a power dynamic that is not encouraged in palliative care as care should be informed by patients’ preferences

Reply: Thank you. We have changed the word ‘compliance’ to ‘adherence’ in the revised manuscript.

Changes in the text: Discussion

Line 275-279: The exact reasons behind the poor symptom control and physical wellbeing were not explored in our study , however the few home-care visits per patient, lack of proper communication between patients and the home care team, lack of necessary skills, insufficient patient and or caregiver adherence, patient education and disease progression might play a role. 

Comment 11: Pg13 line 279-280: It is important to provide balance here as hoping for better outcomes has also been shown to increase uptake of excessive, overaggressive and unnecessary medical interventions towards the end of life which may reduce quality of life.

Reply: Thank you for mentioning this controversy. We have added this issue with relevant references in the discussion part of the revised manuscript.

Changes in the text: Discussion

Line 316-322: A contradictory finding in the current study is about the patients' belief about disease progression. In our study, those who believed the prognosis of the disease as getting worse than before had lower quality of life than those who thought the prognosis was better or the same as before. In several studies, it has been found that conspiracy of silence and false hope lead the patients seeking aggressive but non beneficial treatments. Both of these have a negative consequence on the patients’ quality of life [30-32]. We did not explore the exact reason behind this contradiction. 

Dear Reviewers, we are grateful for your kind time and substantial review; we believe now the manuscript is more improved, which will satisfy you.

---

## [Decision Letter · Decision Letter 1]

3 May 2022

Quality of Life of the Cancer Patients Receiving Home-based Palliative Care in Dhaka City of Bangladesh

PONE-D-21-40845R1

Dear Dr. Biswas,

We’re pleased to inform you that your manuscript has been judged scientifically suitable for publication and will be formally accepted for publication once it meets all outstanding technical requirements.

Kind regards,

Tai-Heng Chen, M.D.

Academic Editor

PLOS ONE

Reviewers' comments:

Reviewer's Responses to Questions

**Comments to the Author**

1. If the authors have adequately addressed your comments raised in a previous round of review and you feel that this manuscript is now acceptable for publication, you may indicate that here to bypass the “Comments to the Author” section, enter your conflict of interest statement in the “Confidential to Editor” section, and submit your "Accept" recommendation.

Reviewer #2: All comments have been addressed

Reviewer #4: All comments have been addressed

2. Is the manuscript technically sound, and do the data support the conclusions?

Reviewer #2: Yes

Reviewer #4: Yes

3. Has the statistical analysis been performed appropriately and rigorously? 

Reviewer #2: I Don't Know

Reviewer #4: Yes

4. Have the authors made all data underlying the findings in their manuscript fully available?

Reviewer #2: Yes

Reviewer #4: Yes

5. Is the manuscript presented in an intelligible fashion and written in standard English?

Reviewer #2: Yes

Reviewer #4: Yes

6. Review Comments to the Author

Reviewer #2: (No Response)

Reviewer #4: Thank you for the hard work in clarifying the methods and findings of the paper. The updated manuscript has satisfactorily addressed my comments.

7. PLOS authors have the option to publish the peer review history of their article (what does this mean?). If published, this will include your full peer review and any attached files.

Reviewer #2: **Yes: **Michael Luis

Reviewer #4: No

---

## [Editor Report · Acceptance letter]

20 Jul 2022

PONE-D-21-40845R1 

Quality of Life of the Cancer Patients Receiving Home-based Palliative Care in Dhaka City of Bangladesh 

Dear Dr. Biswas:

I'm pleased to inform you that your manuscript has been deemed suitable for publication in PLOS ONE. Congratulations! Your manuscript is now with our production department. 

Kind regards, 

on behalf of

Dr. Tai-Heng Chen 

Academic Editor

PLOS ONE